# Evidence against implicit belief processing in a blindfold task

**Katrin Rothmaler**[1,2]*, **Charlotte Grosse Wiesmann**[1]

**1** Minerva Fast Track Research Group Milestones of Early Cognitive Development, Max Planck Institute for Human Cognitive and Brain Sciences, Leipzig, Saxony, Germany, **2** Humboldt Research Group, Faculty of Education, Leipzig University, Leipzig, Saxony, Germany

* rothmaler@cbs.mpg.de

**Data Availability Statement:** The data of all participants who gave informed consent to open access, analysis code, and research materials are available at OSF: https://osf.io/tqy9p/.

## Abstract

Understanding what other people think is crucial to our everyday interactions. We seem to be affected by the perspective of others even in situations where it is irrelevant to us. This intrusion from others' perspectives has been referred to as altercentric bias and has been suggested to reflect implicit belief processing. There is an ongoing debate about how robust such altercentric effects are and whether they indeed reflect true mentalizing or result from simpler, domain-general processes. As a critical test for true mentalizing, the blindfold manipulation has been proposed. That is, participants are familiarized with a blindfold that is either transparent or opaque. When they then observe a person wearing this blindfold, they can only infer what this person can or cannot see based on their knowledge of the blindfold's transparency. Here, we used this blindfold manipulation to test whether participants' reaction times in detecting an object depended on the agent's belief about the object's location, itself manipulated with a blindfold. As a second task, we asked participants to detect where the agent was going to look for the object. Across two experiments with a large participant pool (N = 234) and different settings (online/lab), we found evidence against altercentric biases in participants' response times in detecting the object. We did, however, replicate a well-documented reality congruency effect. When asked to detect the agent's action, in turn, participants were biased by their own knowledge of where the object should be, in line with egocentric biases previously found in false belief reasoning. These findings suggests that altercentric biases do not reflect belief processing but lower-level processes, or alternatively, that implicit belief processing does not occur when the belief needs to be inferred from one's own experience.

## Introduction

As social beings, we strongly depend on the ability to put ourselves in the shoes of others. We excel at understanding what other people think and predicting how they will act. This capacity to reason about the mental states of others is called 'Theory of Mind' (ToM) [1]. We do not use this ability sparingly: we seem to be influenced by the beliefs of other individuals spontaneously, even when they are irrelevant to our current goals. Participants' performance in detecting, recognizing, or counting objects can be impaired, but also improved, when another agent

**Funding:** The authors received no specific funding for this work.

**Competing interests:** The authors have declared that no competing interests exist.

is present who has a different perspective than their own [2–12]. For example, participants are slower to count red dots in a visual scene when an agent is present who can only see some of the dots and, thus, has an incongruent perspective [8]. Conversely, participants are faster to detect a ball that surprisingly appears behind an occluder when a bystander believes the ball to be there [3]. This bias towards the perspective of another person has been referred to as 'altercentric bias' or 'altercentric intrusion' [8, 13]. Recent systematic replication attempts with additional control conditions have revealed a complex picture of successful, partial, and failed replications that triggered a debate about how to interpret altercentric effects and whether they involve true mentalizing or may be explained by simpler processes, like domain-general perceptual processes [2, 14–16].

For visual perspective taking, for instance, a common point of criticism is that the altercentric effects found in the original studies could also be explained by purely perceptual directional cueing [14, 15]. That is, the agent's gaze might simply act as a directional cue that increases participant's visual attention to the agent's visual field. To avoid gaze effects, it was suggested that the agent wear a blindfold that was either transparent or opaque, which participants experienced themselves prior to the experiment [e.g., 17–19]. This blindfold manipulation is considered a critical test for true perspective taking because participants see exactly the same scene, and only their knowledge of the transparency of the blindfold determines what the agent can or cannot see. This blindfold manipulation was, for example, applied to the red dot task described above in several studies and yielded mixed results [20–22]. Some studies indeed only found an altercentric bias when the blindfold led to an incongruent perspective of the agent [20, 21]. This indicates that the bias goes beyond attentional cueing and that the agent's visual perspective, inferred from the blindfold he wore, was taken into account. Other studies, however, reported a bias regardless of the transparency of the blindfold [21, 22], suggesting that participants used the agent's face as a directional cue and did not infer whether he could really see or not based on their own experience with the blindfold.

While an agent's visual perspective is directly visible from a scene and the orientation of the agent's gaze, we also take into account invisible mental states such as people's thoughts or beliefs. This is typically studied with false belief tasks where an agent ends up with a false belief about an object [23, 24]. For instance, in the influential paradigm by Kovács and colleagues [3], a ball first rolls behind an occluder while an agent is watching. The agent then leaves and, while they are away, the ball rolls out from behind the occluder, leaving the scene. The agent thus has a false belief about the ball's location. In a true belief control condition, the agent observes all the ball's movements and thus knows where the ball is. At the end of the trial, the occluder is lowered and participants are asked to press a button if they see a ball behind the occluder. The outcome is incongruent with respect to ball location in half of the trials and thus surprising for the participant. The authors found an altercentric bias in participants' reaction times, i.e., participants were faster to detect an unexpected ball behind the occluder when the agent falsely believed the ball to be there, compared to when the agent also knew that the ball was gone. A critical question is whether participants will also show such altercentric bias when the belief of the agent is manipulated with a blindfold, rather than their presence or absence. This would allow for false belief and true belief conditions to be identical, except for participants' knowledge concerning the visibility through the blindfold. It would thus exclude the possibility that the altercentric bias resulted from the fact that the agent's presence or absence, during the location change, impacted the saliency of the event and, thus, the participants' memory of the actual object location. In the current study, we therefore tested whether adults show an altercentric bias in a blindfold false belief task, that is, in a setting where the agent's belief could only be deduced from the participant's own experience with the transparency of a blindfold.

While another person's perspective can affect our own behavior, it can also be the other way around. That is, our own perspective can affect our ability to understand what other people think. For instance, participants were slower and less accurate in their conclusions about what another person could see or believed when their own perspective did not match that of the other person [8, 25]. This bias towards one's own perspective has been referred to as 'egocentric bias' [8, 25]. Since it impedes the ability to infer others' perspectives, the egocentric bias can be considered a negative measure of ToM reasoning. Thus, there might be a negative correlation between altercentric and egocentric biases. That is, people who are very receptive to the perspective of others are less likely to be impaired by their own perspective. Therefore, a second aim of the present study was to investigate the relation between altercentric and egocentric biases in response times within the same experimental setting.

To test for an altercentric bias in response times, we deployed an *object detection task* that was inspired by the above mentioned paradigm of Kovács and colleagues [3]. There has been some debate about whether their observed differences in reaction time might have been due to timing confounds [16]. However, El Kaddouri et al. [2] recently replicated the original findings using a new adjusted version of the task in which they equalized the timing in all conditions. In contrast to the original paradigm, however, in the current study, we manipulated the belief of the agent with a blindfold. Manipulating the belief in this way has the advantage that the stimuli are exactly the same for both experimental conditions. Their interpretation changes only because of the participants' own experience with the blindfold. If altercentric interference occurred in this setting, this would strengthen the conclusion that altercentric bias results from true mentalizing and not from attention or memory effects depending on the presence or absence of an agent. An opaque blindfold would result in a false belief of the agent about the object's location, predicting that participants should be faster in detecting the object in the unexpected but belief-congruent location, as in the previous studies without a blindfold [2, 3, 16]. Conversely, participants may be slowed down in detecting the ball when the agent believed the object to be in the other location.

To be able to study altercentric and egocentric interferences within the same experimental setting, after the object detection task, we asked participants to perform an *action detection task*. Instead of detecting the object behind the occluder, in this task, participants were asked to detect where the agent would look for the object. As in the object detection task, the outcome was surprising for the participants in half of the action detection trials. That is, the agent did not always look for the object where she believed it to be. If participants engaged in ToM and predicted the agent's action based on her belief about the object's location, they should be slower in detecting such incongruent behavior. We predicted that when the agent held a true belief about the object's location, participants' responses should be faster if she looked for it in the correct location. In contrast, when the agent held a false belief, they should be faster to detect her action if she looked in the incorrect, but believed, location. If participants experienced an egocentric bias during this task, their response times should also be affected by their own belief about the object's location. That is, participants responses should be faster when the agent looked for the object in its actual location, regardless of the agent's belief.

We carried out two preregistered experiments using the same stimuli but different experimental setups. Experiment 1 was conducted online, experiment 2 in the laboratory.

## Experiment 1 –online

Experiment 1 consisted of an online study that was preregistered at https://aspredicted.org/MEC_TTM. All analyses were conducted according to the preregistration unless explicitly stated otherwise.

## Material and methods

**Participants.**   N = 119 German native speakers, without current or prior neurological or psychological disorders, participated voluntarily in the online experiment (40 male, 79 female, aged between 18 and 58, mean age = 27.59, median = 25). From these 119 participants, 90 could be included into the analyses for the object detection task (altercentric bias), and 102 for the action detection task (egocentric bias). Participants provided informed consent and were compensated monetarily for their time. The study was approved by the ethics committee at the Medical Faculty of Leipzig University.

**Stimuli and task.**   Participants watched a series of short movie clips featuring an agent observing a scene on a table ([Fig 1]). This scene consisted of two boxes and an object that entered the scene and moved into one of the boxes. While the object stayed in the box, the agent put on a blindfold that was either transparent or opaque, which participants had to deduce from their own knowledge of the blindfold. The object then moved from one box to the other, leading to either a true belief (TB, transparent blindfold) or a false belief (FB, opaque blindfold) of the agent concerning the object's location. Participants had two different tasks: an object or an action detection task. The object detection task was conducted to access participants' altercentric bias as a measure of their implicit ToM performance, while the action detection task aimed to detect an egocentric bias as a possible negative measure of their explicit ToM performance (action detection). The end of the movie depended on these tasks. In the object detection task, the boxes moved aside and revealed the location of the object. In half of the movies, the object appeared where the participants expected it to be (reality congruent, RC). In the other half, it appeared at the opposite side (reality incongruent, RI). Participants had to press the right button as fast as possible if they detected the object on the right side of

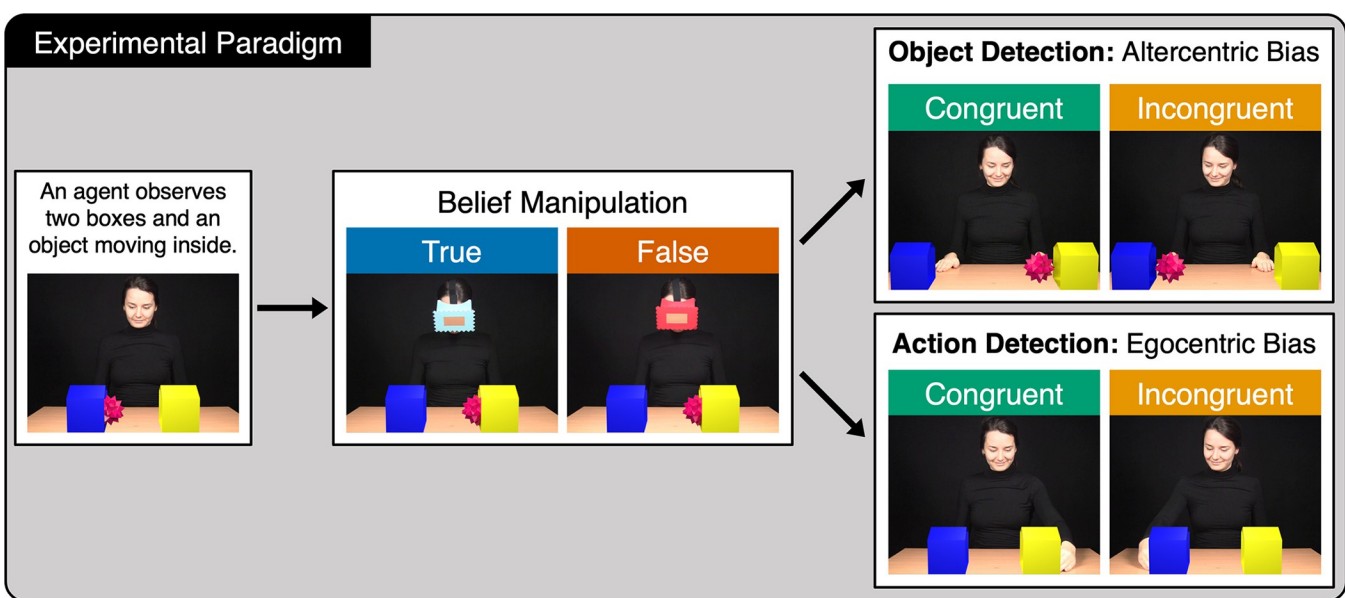

**Fig 1. Overview of the experimental paradigm.** An agent observes an object moving into one of two boxes. She then puts on a mask that is either transparent (true belief condition) or opaque (false belief condition, colors counterbalanced across participants). The participant is familiarized with the transparency of the blindfolds prior to the experiment. After the agent put on the mask, the object changes from one box to the other. The participant then has two different tasks. In the object detection task, the boxes move aside at the end of the video revealing the object either where the participant last saw it (reality congruent) or at the opposite side of the screen (reality incongruent). Participants need to detect the object by pressing the button on the corresponding side as soon as possible. In the action detection task, the agent reaches either into the box where she believes the object to be or into the box that she believes to be empty. Her facial expression remains the same in both conditions. Participants need to detect her grasping movement as soon as possible.

the screen, and the left button if they detected it on the left. We used a fully-crossed design, with left and right button presses counterbalanced across conditions. For the action detection task, the boxes did not move at the end of the movie to reveal the object, but instead, the agent reached into one of the boxes. In half of the movies, she reached into the box where she believed the object to be. In the other half, she reached into box she believed to be empty. Participants were instructed to detect the agent's action based on her belief and press the right button as fast as possible if they detected the agent reaching toward the right box and the left button if they detected her reaching toward the left box. The movie clips thus differed with respect to the following four factors: outcome (object vs. action detection task), belief (TB vs. FB), reality congruency (RC vs. RI) and button press side (right vs. left). Fig 1 provides an overview of the experimental paradigm. The individual in this figure has given written informed consent (as outlined in the PLOS ONE consent form) to publish her photo. An exemplary video of the action detection task can be found here: www.cbs.mpg.de/milestones/video-to1beli.

**Procedure.** Prior to the experiment, participants were familiarized with the opaque and the transparent blindfold. They learned about their transparency in written form and had to successfully infer at least three times what an agent wearing those blindfolds could or could not see. Afterwards, they completed eight object detection trials and, after a brief break, eight action detection trials. Since the action detection task draws participants' attention to the belief of the agent and, thus, potentially influences implicit altercentric biases in the object detection task, the object detection task was always conducted first. Each task block was preceded by two training videos plus control question about the transparency of the blindfolds (i.e., 'Could the agent see through the blindfold?') that were repeated until participants answered both right. Within the experimental blocks, the trials were shuffled randomly while making sure that the trials required for the calculation of the main contrasts remained close. The participants' attention to the blindfolds was checked by four control questions (i.e., 'Could the agent see through the blindfold?') that were posed between the trials.

**Data collection and preprocessing.** Button presses were recorded via the online platform Pavlovia. For object detection trials, only button presses were included that occurred after the first bit of the object became visible. Reaction times (RTs) were calculated relative to this time point. For action detection trials, RTs were calculated relative to the agent's first eye movement, which indicated the direction of her action, and only button presses after this time point were included in the analysis. Responses that were incorrect, that were given too late (after the curtain indicating the end of the trial), that followed any technical errors, or that were more than three standard deviations above or below the grand average were excluded. Following our preregistration, participants were excluded from the analysis of a task if they showed non-meaningful task-compliance (i.e., more than 25% errors or more than one incorrect control question per task), if their reaction times differed substantially from the rest of the sample (i.e., more than three standard deviations above or below the grand average), or if they reported any history of neurological or psychiatric disorders. In addition to those preregistered criteria, we excluded participants who experienced technical problems, who did not fully understand the task instructions according to self-report or gave consistent incorrect responses in one condition, and who provided less than 6 out of total 16 trials for analysis. The latter criterium was added to make sure that exclusion of individual responses did not leave a participant with an insufficient amount of data for the analysis.

As preregistered, we log-transformed our data, as is common for non-normal reaction times, since the quantile-quantile (Q-Q) plots indicated right skewed distributions. After log-transformation only minor deviations from normality were observed (see S1 Fig in S1 File). Thus, we decided to move forward with our original analysis plan as all conducted analyses are fairly robust to violations of the normality assumption [26].

**Bayesian framework.** We preregistered a Bayesian framework for all of our analyses since it allows sequential testing and can gather evidence for the presence as well as absence of an effect. Bayesian statistics rely on model comparisons. For each model and all parameters involved, prior probability distributions need to be specified [27]. Since we had no a priori information about our effects, we gave all models an equal, uninformed prior probability that is also called *default prior* [28]. Once the prior distributions are specified, each model's marginal likelihood, given the observed data, can be calculated and contrasted using the Bayes Factor (BF). The $BF_{10}$ sets the evidence for the alternate model (denoted with subscript 1) in contrast to the evidence for the null model (denoted with subscript 0). Although the BF assesses evidence on a continuous scale, there is a widely accepted classification scheme. Evidence for the alternate model is considered anecdotal for a $BF_{10}$ between 1 and 3, moderate between 3 and 10, strong when greater than 10, very strong when greater than 30, and extreme for a $BF_{10}$ greater than 100. Likewise, evidence for the null model is considered anecdotal for a $BF_{10}$ between 1 and 1/3, moderate between 1/3 and 1/10, strong when smaller than 1/10, very strong when smaller than 1/30, and extreme for a $BF_{10}$ smaller than 1/100. All Bayesian analyses were conducted using the R package BayesFactor version 0.9.12–4.2 [29] or JASP (Version 0.16).

**Sequential testing.** Bayesian Statistics allows for sequential testing, i.e., testing can be continued until the Bayes Factor of the predefined effect of interest reaches one of two predefined, symmetric thresholds. In the present case, we preregistered the interaction between the factor *reality congruency* and the factor *belief* as our effect of interest as this interaction quantifies evidence for the presence of an altercentric or egocentric bias, respectively and a BF of 4 or ¼ as threshold for stopping data collection. To control for false positive and false negative results, data from a minimum of 68 participants was collected. This minimal number was determined by means of power simulations using the R packages 'Superpower' version 0.1.0 [30] and 'BayesFactor' version 0.9.12–4.2 (Morey & Rouder, 2015). More details about power simulations and sequential testing as well as the convergence plots can be found in the supplements (sections *S1 Power Simulations* in S1 File and *S3 Experiment 1*: *Sequential Testing* in S1 File).

**Transparency and openness.** In the above sections, we report how we determined our sample size, all data exclusions, all manipulations, and all measures in the study. The data of all participants who gave informed consent to open access, analysis code, and research materials are available at OSF: https://osf.io/tqy9p. Data were analyzed using Matlab version R2020b, Rstudio version 2022.02.3, and JASP version 0.16. This study's design, its analysis and hypotheses were pre-registered at AsPredicted (https://aspredicted.org/MEC_TTM).

## Results

**Object detection task.** To test if the reaction times in the object detection task were modulated by the belief of the agent or the expectation of the participant, we computed a linear mixed model (LMM) with the individual reaction times (IRTs) as dependent variable, *belief* and *reality congruency* as independent variables and a random intercept per subject. This model provided extreme evidence for a main effect of *reality congruency* ($BF_{10} > 10,000$, see Table 1 for all Bayes Factors and model comparisons). Participants responded on average faster if the object appeared at the expected (i.e., reality congruent) compared to the unexpected (i.e., reality incongruent) location (RC: Avg = 892 ms, SD = 190 ms, SE = 20 ms, RI: Avg = 939 ms, SD = 207 ms, SE = 21 ms, see also Fig 2 left). For the factor *belief*, our model comparisons yielded strong evidence against a main effect ($BF_{10} = 0.09$). Also, contrary to our hypothesis, we found moderate evidence against an interaction between *reality congruency* and *belief* ($BF_{10} = 0.14$). As previous research had mostly found a difference in reaction times

**Table 1. Bayes factors for the model comparisons in experiment 1.**

| | Model 0 (1st row) and Model 1 (2nd row) | Object Detection Task | | | | Action Detection Task | | | |
|---|---|---|---|---|---|---|---|---|---|
| | | BF | Error | BF10 | Error | BF | Error | BF10 | Error |
| Belief | IRT ~ ID + C | 15927.8200 | 0.29% | 0.0863 | 1.28% | 0.2907 | 0.67% | 0.1360 | 0.83% |
| | IRT ~ ID + C + B | 1374.1380 | 1.24% | | | 0.0395 | 0.48% | | |
| Congruency | IRT ~ ID + B | 0.0853 | 0.50% | 16112.17 | 1.34% | 0.1369 | 0.33% | 0.2886 | 0.58% |
| | IRT ~ ID + B + C | 1374.1380 | 1.24% | | | 0.0395 | 0.48% | | |
| Belief * Congruency | IRT ~ ID + B + C | 1374.1380 | 1.24% | 0.1389 | 1.53% | 0.0395 | 0.48% | 9.21 | 0.97% |
| | IRT ~ ID + B + C + B*C | 190.8933 | 0.89% | | | 0.3640 | 0.85% | | |
| Belief | IRT ~ ID + O + BP + C | 6.23E+128 | 2.45% | 0.0881 | 1.91% | 7.94E+189 | 1.85% | 0.1225 | 2.17% |
| | IRT ~ ID + O + BP + C + B | 4.99E+134 | 1.59% | | | 9.73E+188 | 1.12% | | |
| Congruency | IRT ~ ID + O + BP + B | 3.61E+129 | 2.46% | 138257.30 | 2.93% | 5.10E+189 | 1.52% | 0.1907 | 1.89% |
| | IRT ~ ID + O + BP + B + C | 4.99E+134 | 1.59% | | | 9.73E+188 | 1.12% | | |
| Order | IRT ~ ID + BP + B + C | 6.23E+128 | 2.45% | 801253.30 | 2.92% | 7.04E+183 | 1.09% | 138125.60 | 1.56% |
| | IRT ~ ID + O + BP + B + C | 4.99E+134 | 1.59% | | | 9.73E+188 | 1.12% | | |
| Button Press | IRT ~ ID + O + B + C | 8.59E+134 | 1.45% | 0.5814 | 2.15% | 1.25E+190 | 1.85% | 0.0776 | 2.16% |
| | IRT ~ ID + O + BP + B + C | 4.99E+134 | 1.59% | | | 9.73E+188 | 1.12% | | |
| Order*Belief | IRT ~ ID + O + BP + B + C + C*O + B*C | 4.58E+142 | 2.11% | 0.1506 | 2.53% | 1.43E+190 | 2.20% | 0.3272 | 2.81% |
| | IRT ~ ID + O + BP + B + C + C*O + B*C + B*O | 6.90E+141 | 1.39% | | | 4.69E+189 | 1.75% | | |
| Order * Congruency | IRT ~ ID + O + BP + B + C + B*O + B*C | 6.93E+132 | 1.92% | 9.96E+08 | 2.37% | 3.03E+189 | 1.81% | 1.55 | 2.52% |
| | IRT ~ ID + O + BP + B + C + B*O + B*C + C*O | 6.90E+141 | 1.39% | | | 4.69E+189 | 1.75% | | |
| Belief * Congruency | IRT ~ ID + O + BP + B + C + C*O + B*O | 5.88E+142 | 1.64% | 0.1173 | 2.15% | 3.53E+188 | 1.57% | 13.29 | 2.35% |
| | IRT ~ ID + O + BP + B + C + C*O + B*O + B*C | 6.90E+141 | 1.39% | | | 4.69E+189 | 1.75% | | |

Table 1 shows the null model (Model 0) and the alternative model (Model 1) for the different model comparisons along with the respective Bayes Factors of the models (BF), their errors, as well as the Bayes Factors of the model comparisons (BF$_{10}$) with their respective errors. Abbreviations: IRT = individual reaction times, ID = subject intercept, C = Reality Congruency, B = Belief, BP = Button Press Side, O = Trial Order

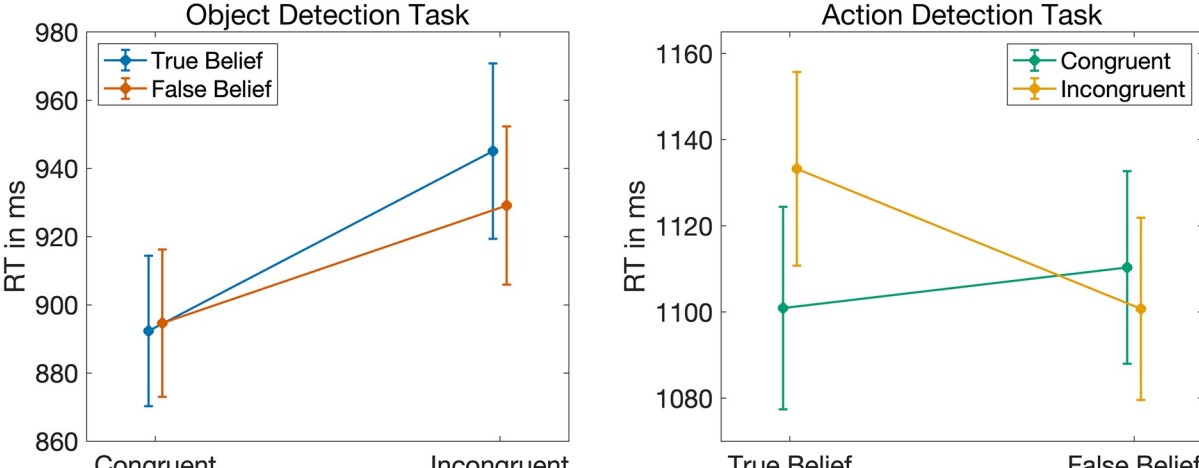

**Fig 2. Response times in experiment 1.** Left: Grand averages of the reaction times in the object detection task along with standard errors for true belief trials (blue) and false belief trials (red) as a function of reality congruency. Right: Grand averages of the reaction times in the action detection task along with standard errors for reality congruent trials (green) and reality incongruent trials (orange) as a function of belief.

**Table 2. Bayes factors for the pairwise comparisons.**

| Task | Condition | Comparison | BF10 | Error in % |
|------|-----------|-----------|------|-----------|
| Object Detection | RI | FB<TB | 0.149 | 4.15E-05 |
|  | RC | TB<FB | 0.1999 | 3.37E-05 |
| Action Detection | FB | RI<RC | 0.322 | 1.76E-04 |
|  | TB | RC<RI | 23.497 | 2.85E-04 |

Table 2 shows the Bayes factors along with the corresponding errors for the pairwise comparisons in the object detection and the action detection tasks for reality incongruent trials (RI, preregistered direction: reaction times in false belief trials were shorter than in true belief trials) and for reality congruent trials (RC, preregistered direction: reaction times in true belief trials were shorter than in false belief trials).

for reality incongruent trials, depending on the belief of the agent [2, 3, 16], we also contrasted the mean reaction times in false belief and true belief reality incongruent trials of the object detection task using a Bayesian paired directed t-test (FB < TB, planned comparison). Unlike previous research without a blindfold design, this planned comparison yielded moderate evidence against participants being faster when the agent had a false belief, compared to when the agent had a true belief about the object's location in reality incongruent trials ($BF_{10} = 0.1$, error = 0.0001%, see Table 2). Similarly, we found moderate evidence against participants being faster in true belief than in false belief reality congruent trials ($BF_{10} = 0.2$, error < 0.0001%, see Table 2).

In a second step, we incorporated *button press side* and *trial order* as covariates in the model. Since there could be an interaction between *trial order* and *reality congruency* as well as *trial order* and *belief*, we included those interactions in the model. Including these covariates did not alter the results (reality congruency: $BF_{10} > 100,000$, belief: $BF_{10} = 0.09$, interaction: $BF_{10} = 0.12$, see Table 1). There was extreme evidence for a main effect of *trial order* ($BF_{10} > 100,000$) reflecting the fact that participants became faster over time. For *button press side*, evidence remained ambiguous (BF = 0.58). The evidence for an interaction between *trial order* and *reality congruency* was extreme ($BF_{10} > 1,000,000$). That is, there was a stable decrease of the average reaction times over the course of the experiment for reality incongruent trials, but the opposite was the case for reality congruent trials, wherein participants adapted to the occurrence of reality incongruent trials (see Fig 3). This adjustment occurred more quickly when the agent had a false belief compared to when the agent had a true belief about the object's location. To be more precise, there was evidence for a reality congruency effect in the first two trials when the agent had a true belief (paired directed Bayesian t-test comparing RC < RI for first trials only: $BF_{10} = 67$, analogous test for second trials only: $BF_{10} = 3.6$). When the agent had a false belief, in contrast, there was only evidence for a reality congruency effect in the first trial (paired directed Bayesian t-test comparing RC < RI for first trials only: $BF_{10} = 67$) but not in the second (analogous test for second trials only: $BF_{10} = 0.26$). Finally, the model comparisons provided moderate evidence against an interaction between *trial order* and *belief* ($BF_{10} = 0.15$).

**Action detection task.**   To test if the reaction times in the action detection task were modulated by the belief of the agent or the expectation of the participant, we computed a linear mixed model (LMM) with the individual reaction times (IRTs) as dependent variable, *belief* and *reality congruency* as independent variables and a random intercept per subject. This model yielded moderate evidence against the two main effects (reality congruency: $BF_{10} = 0.29$, belief: $BF_{10} = 0.14$, see Table 1 for all Bayes Factors and model comparisons), but moderate evidence for an interaction between *reality congruency* and *belief* ($BF_{10} = 9$) as hypothesized. That is, participants responded faster in true belief trials if the agent reached into the

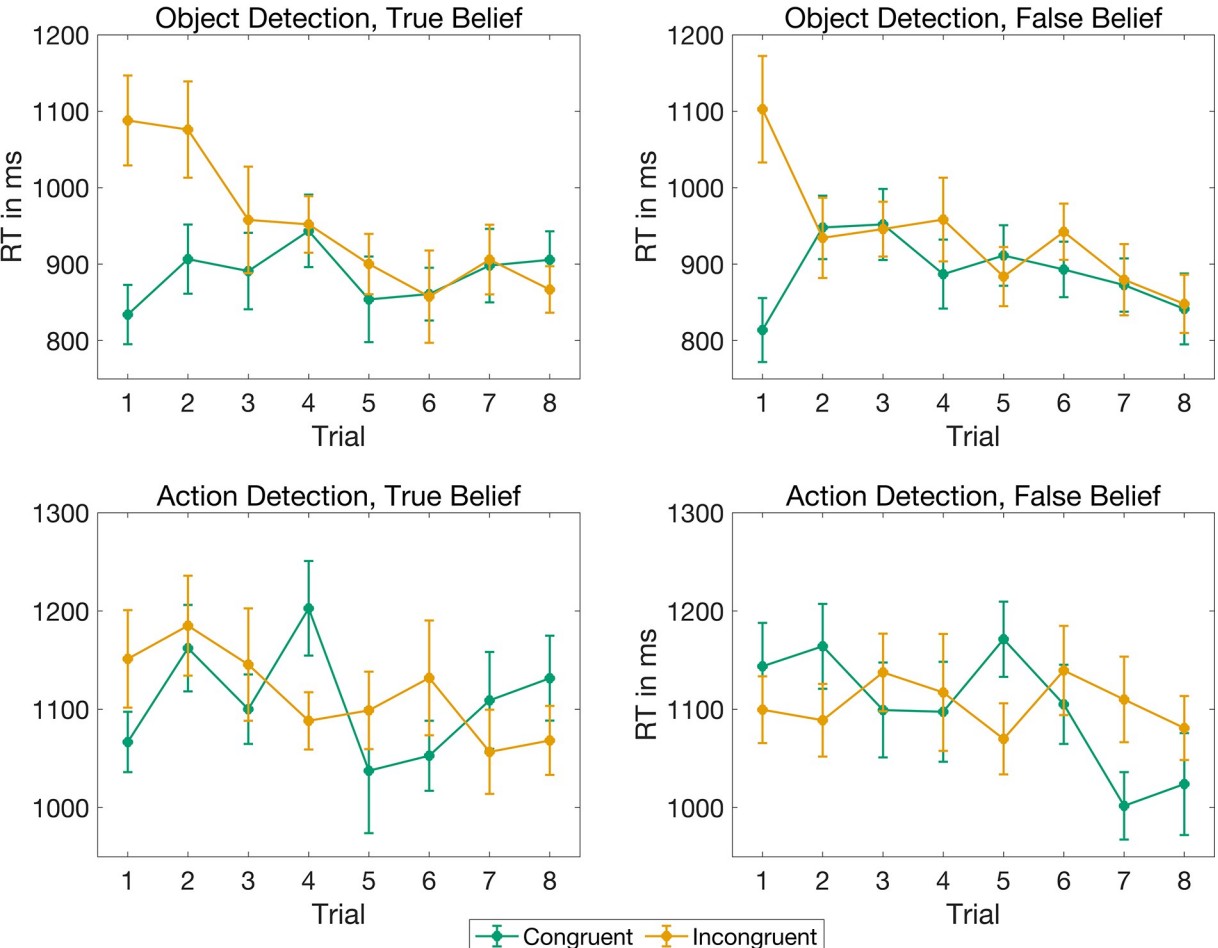

**Fig 3. Response times in the object and action detection task as a function of trial number.** Grand averages of the response times in the object detection task (upper panel) and in the action detection task (lower panel) for true belief trials (left) and false belief trials (right) along with standard errors for reality congruent (green) and reality incongruent trials (orange) as a function of trial number.

box where the object actually was (i.e., reality congruent) instead of reaching into the empty box (i.e., reality incongruent) (RC: Avg = 1103 ms, SD = 228 ms, SE = 23 ms, RI: Avg = 1136 ms, SD = 214 ms, SE = 21 ms, see also Fig 2 right graphic). In false belief trials, this pattern was reversed. That is, participants reacted faster if the agent reached into the empty box (which was the box where she actually believed the object to be) instead of reaching into the box where the object actually was (RC: Avg = 1118 ms, SD = 225 ms, SE = 22 ms, RI: Avg = 1105 ms, SD = 207 ms, SE = 21 ms). It needs to be noted that, in this particular analysis, the Bayes Factors of the single models entering the model comparisons were very small (see Table 1). If the Bayes Factors of the single models are small, there is little evidence for the models themselves and the results of the model comparison need to be interpreted with caution. However, when including the covariates *trial order* and *button press side*, the Bayes Factors of the single models became very large and the results of the model comparisons remained similar (reality congruency: $BF_{10} = 0.19$, belief: $BF_{10} = 0.12$, interaction: $BF_{10} = 13$), increasing confidence in the findings.

For the action detection task, we preregistered two planned comparisons. First, we wanted to compare the reaction times in reality incongruent and reality congruent true belief trials

using a Bayesian paired directed t-test (RC < RI). This t-test revealed strong evidence for an effect of reality congruency ($BF_{10}$ = 24, error < 0.0003%, see Table 2). Second, we wanted to compare the reaction times in false belief trials using the same test with the opposite direction (RI < RC). This test yielded moderate evidence against participants being faster when the agent acted according to her belief compared to when she acted against it, reflecting an egocentric bias in response times (directed Bayesian t-test: $BF_{10}$ = 0.32, error < 0.0002%). As in the object detection task, there was extreme evidence for a main effect of *trial order* ($BF_{10}$ > 100,000). That is, participants became faster over time. For *button press side*, the model comparison yielded strong evidence against a main effect ($BF_{10}$ = 0.08) and there was moderate evidence against an interaction between *trial order* and *belief* ($BF_{10}$ = 0.33), while the evidence for an interaction between *trial order* and *reality congruency* remained inconclusive ($BF_{10}$ = 1.55).

**Relation between altercentric and egocentric biases.** In addition to our preregistered analyses, we examined the relation between the altercentric and the egocentric bias with a Bayesian correlation analysis. For this analysis, the altercentric bias was defined according to the critical comparison of Kovács et al. [3], i.e., as the mean response time in true belief reality incongruent trials minus the mean response time in false belief reality incongruent trials. If participants benefited from the false belief of the agent, this measure should be greater than zero. Analogously, the egocentric bias was defined as the mean response time in reality incongruent false belief trials minus the mean response time in reality congruent false belief trials. If reaction times were only modulated by the belief of the agent but not by the belief of the participant (i.e., no egocentric bias but perfect explicit ToM performance), this measure would be smaller than zero. If, however, participants exhibited an egocentric bias, this measure would collapse to zero or even become positive. Since the egocentric bias can be interpreted as a negative measure of ToM and the altercentric bias as a positive indicator of ToM, we expected a negative correlation between the two. However, this analysis yielded moderate evidence against a negative correlation between the altercentric and the egocentric bias (Pearson's r = -0.032, $BF_{10}$ = 0.171).

## Discussion experiment 1 (online)

Previous studies provided evidence for an altercentric bias in reaction times in an object detection false belief task. That is, participants were faster in detecting a ball that surprisingly appeared at a certain location when an agent expected the ball to be there compared to when they did not [2, 3, 16]. To test if such altercentric biases result from true mentalizing, the blindfold manipulation was proposed. Using this blindfold manipulation, we found moderate evidence against an altercentric effect, i.e., participants did not benefit from the false belief of the agent when detecting an object that appeared at an unexpected location. Moreover, our model comparisons provided strong evidence against a main effect of *belief* and moderate evidence against an interaction between *belief* and *reality congruency*. This indicates that participants did not process the belief of the agent in the object detection task. We did find, however, extreme evidence for an effect of *reality congruency* as hypothesized. That is, participants responded faster if the object appeared at the expected location.

For the action detection task, we found evidence for the expected egocentric bias in participants' response times. That is, we found moderate evidence for an interaction between *reality congruency* and *belief*. While there was an effect of *reality congruency* in the true belief condition, which could reflect both belief processing as well as egocentric interference, we found evidence against such an effect in the false belief condition. In the false belief condition, the agent acted according to her (false) belief when she reached for the reality incongruent location. Thus, if participants' response times were only guided by the agent's belief, they should have

been faster in reality incongruent than in reality congruent trials. The absence of such an effect therefore indicates the presence of an egocentric bias in response times.

In both, the object and the action detection task, we found extreme evidence for a main effect of *trial order*. This effect reflects learning processes, i.e., participants became faster during the course of the experiment. Further, after they had experienced that the object could also be revealed in the reality incongruent location in the object detection task, participants became faster on subsequent incongruent and slower on subsequent congruent trials, reflecting their changed expectations about the possible object locations (for an illustration see Fig 3). Interestingly, participants adjusted their expectations faster when the agent had a false belief compared to when the agent had a true belief about the object's location. In the action detection task, the response pattern in the first two trials indicates that participants' responses were initially mainly driven by the belief of the agent, i.e., they were faster when the agent acted according to her belief. Over the course of the experiment, however, their own knowledge about the object location became dominant in false belief trials.

Finally, we found moderate evidence against a correlation between the altercentric and the egocentric bias. However, the lack of a relation between these measures of implicit and explicit ToM may be due to the lack of an altercentric bias and should be interpreted with caution.

Taken together, we did not find the expected altercentric bias in response times in experiment 1. There are several possible explanations for this. First of all, the precision of a button press recorded online is much lower than in the lab, and the attention span and thus number of trials is a lot lower. It is therefore possible that very small effects may not have been detected. Moreover, participants learned about the transparency of the glasses only in written form and not by firsthand experience. Although they answered several control questions about what an agent wearing those glasses could or could not see correctly, trying the glasses themselves might help participants distinguish between the two more intuitively. To rule out these possibilities, we conducted a second experiment in the lab, in which participants could gain firsthand experience with the glasses.

## Experiment 2 –laboratory

Experiment 2 was conducted in the lab. Stimuli, data analysis software, open data policies and Bayesian Sequential Testing scheme (i.e., interaction between the factor *reality congruency* and the factor *belief* as effects of interest and thresholds of 4 and ¼) were the same as in experiment 1. The convergence plots and more details about the sequential testing can be found in the supplements (see S5 Fig in S1 File). The study was preregistered at https://aspredicted.org/blind. php?x=ZKS_OES.

### Materials and methods

**Participants.**   N = 115 healthy, right-handed German native speakers participated voluntarily in the lab experiment (53 male, 62 female, aged between 18 and 38, mean age = 25.14, median = 25). Of these 115 participants, 108 were included in the analysis for the object detection task (altercentric bias) and 110 in the action detection task (egocentric bias). Inclusion criteria, informed consent, monetary compensation, and ethics approval were analogous to experiment 1.

**Procedure.**   Participants completed 64 object detection trials and, after a short break, 64 action detection trials. As in experiment 1, we wanted to assess altercentric bias in response times in the object detection task as a measure of implicit ToM and egocentric bias in the action detection task as a negative measure of explicit ToM. To ensure that potential implicit altercentric bias in the object detection task were not influenced by any belief processing

during the action detection task, the object detection task was, again, always conducted first. In the lab, the belief manipulation (TB vs. FB) was presented in blocks (TB and FB order counterbalanced across participants), as pilot studies showed that it was difficult for participants to keep track of the blindfolds over a long period of time (the lab experiment lasted about 1h). Before each block, only the blindfold that the agent would wear in the upcoming block was introduced. Participants were asked to put on this blindfold, and thus, learned about their transparency through experience. In addition, they had to successfully infer what an agent wearing those blindfolds could or could not see, solving two training trials correctly and answering one final control question (i.e., 'Could the agent see through the blindfold?'). Only participants who answered these control questions correctly were included (n = 6 were excluded from the object detection task, n = 0 for the action detection task). Within the four blocks, the trials were shuffled randomly while making sure that the first two trials of each task included a reality congruent and a reality incongruent trial with button press on the same side. This was done to assure that we could compare the first two trials in the lab with the results of the online experiment, and because participants' expectations strongly depended on the first occurrence of such a trial.

**Data collection and preprocessing.** Button presses were recorded with the Neurobehavioral Systems Inc. software Presentation. Exclusion criteria were analogous to experiment 1. Depending on the analysis either average RTs (ARTs, average of all valid trials within each experimental condition) or IRTs were utilized as the dependent variable. As preregistered and analogously to experiment 1, we log-transformed our data since Q-Q plots of both ARTs and IRTs indicated right skewed distributions that deviated from normality. We still observed some deviations from normality for the log-transformed data, but these were mainly driven by a few outliers (see S3 and S4 Figs in S1 File). To assure that these outliers did not drive our results, we repeated all analyses after removing elements that were more than 1.5 interquartile ranges above the upper or below the lower quartile. This yielded similar results as when these values were included (see section *S5 Experiment 2*: *Violation of normality* in S1 File for more details).

## Results

**Object detection task–first trials.** To be able to compare the results of experiment 1 and 2, in a first analysis, we only included the first two object detection trials of every participant. To test for an altercentric bias in reaction times, we analyzed these individual response times using a linear mixed model including the within-subject factor *reality congruency*, and the between-subject factor *belief*. This analysis provided extreme evidence for a main effect of *reality congruency* ($BF_{10} > 1,000,000$, see Table 3 for all Bayes Factors and model comparisons). As expected, participants were faster if the object appeared at the expected (i.e., reality congruent) location compared to the unexpected (i.e., reality incongruent) location (RC: Avg = 460ms, SD = 147 ms, SE = 14 ms, RI: Avg = 634 ms, SD = 163 ms, SE = 18 ms, see also Fig 4 upper left). For the factor *belief*, the model comparison provided moderate evidence against a main effect ($BF_{10} = 0.29$), while the Bayes Factor of the interaction between *belief* and *reality congruency* remained inconclusive ($BF_{10} = 1$). As in experiment 1 and analogous to previous research on altercentric biases in object detection [2, 3, 16], we also directly compared the reaction times in false belief and true belief reality incongruent trials using a Bayesian paired directed t-test (FB < TB). In accordance with the results of our first experiment, but in contrast to previous findings without the blindfold [2, 3, 16], this planned comparison provided moderate evidence against an altercentric effect, i.e., against participants being faster in reality incongruent trials when the agent had a false belief compared to when the agent had a

**Table 3. Bayes factors for the model comparisons in experiment 2.**

| | | | Object Detection Task | | | | Action Detection Task | | | |
|---|---|---|---|---|---|---|---|---|---|---|
| | | **Model 0 (1st row) and Model 1 (2nd row)** | **BF** | **Error** | **BF10** | **Error** | **BF** | **Error** | **BF10** | **Error** |
| First Trials | Belief | IRT ~ ID + C | 3.21E+13 | 0.53% | 0.2896 | 0.61% | 0.1933 | 0.29% | 0.2237 | 0.89% |
| | | IRT ~ ID + C + B | 9.31E+12 | 0.32% | | | 0.0432 | 0.85% | | |
| | Congruency | IRT ~ ID + B | 0.2555 | 0.32% | 3.64E+13 | 0.45% | 0.2182 | 0.61% | 0.1982 | 1.04% |
| | | IRT ~ ID + B + C | 9.31E+12 | 0.32% | | | 0.0432 | 0.85% | | |
| | Belief * Congruency | IRT ~ ID + B + C | 9.31E+12 | 0.32% | 1.0369 | 1.06% | 0.0432 | 0.85% | 386.9398 | 1.58% |
| | | IRT ~ ID + B + C + B*C | 9.65E+12 | 1.01% | | | 16.7315 | 1.33% | | |
| All Trials | Belief | ART ~ ID + C + IB | 19.2144 | 0.94% | 0.1443 | 2.07% | 1.3064 | 0.56% | 0.1790 | 0.90% |
| | | ART ~ ID + C + IB + B | 2.7725 | 1.85% | | | 0.2339 | 0.70% | | |
| | Congruency | ART ~ ID + B + IB | 0.5908 | 0.42% | 4.6931 | 1.89% | 0.0741 | 0.58% | 3.1583 | 0.91% |
| | | ART ~ ID + B + IB + C | 2.7725 | 1.85% | | | 0.2339 | 0.70% | | |
| | Initial-Belief | ART ~ ID + B + C | 0.7220 | 0.48% | 3.8399 | 1.91% | 0.5489 | 1.46% | 0.4261 | 1.62% |
| | | ART ~ ID + B + C + IB | 2.7725 | 1.85% | | | 0.2339 | 0.70% | | |
| | Initial-Belief * Belief | ART ~ ID + B + C + IB + C*IB + B*C | 0.0651 | 1.10% | 9.55E+36 | 4.07% | 0.0129 | 1.06% | 3.76E+26 | 1.73% |
| | | ART ~ ID + B + C + IB + C*IB + B*C + B*IB | 6.21E+35 | 3.92% | | | 4.86E+24 | 1.37% | | |
| | Initial-Belief * Congruency | ART ~ ID + B + C + IB + B*IB + B*C | 4.73E+36 | 2.73% | 0.1314 | 4.77% | 3.02E+25 | 1.16% | 0.1609 | 1.80% |
| | | ART ~ ID + B + C + IB + B*IB + B*C + C*IB | 6.21E+35 | 3.92% | | | 4.86E+24 | 1.37% | | |
| | Belief * Congruency | ART ~ ID + B + C + IB + B*IB + C*IB | 4.63E+36 | 2.34% | 0.1342 | 4.56% | 9.46E+24 | 1.14% | 0.5137 | 1.79% |
| | | ART ~ ID + B + C + IB + B*IB + C*IB + B*C | 6.21E+35 | 3.92% | | | 4.86E+24 | 1.37% | | |

Table 3 shows the null model (Model 0) and the alternative model (Model 1) for the different model comparisons along with the respective Bayes Factors of the models (BF) and their errors as well as the Bayes Factors of the model comparisons (BF$_{10}$) with their respective errors. Abbreviations: IRT = individual reaction times, ART = average reaction times, ID = subject intercept, C = Reality Congruency, B = Belief, BP = Button Press Side, O = Trial Order, IB = Initial Belief

true belief (BF$_{10}$ = 0.1, error = 0.002%, see Table 4). Similarly, a comparison including only the data of reality congruent trials yielded moderate evidence against the hypothesis that participants were faster when the agent had a true and not a false belief about the object's location (BF$_{10}$ = 0.1, error = 0.001%).

**Object detection task–all trials.** In a second analysis we included all response times that were collected during the lab experiment. Average reaction times were analyzed with a repeated-measures ANOVA including the two within-subject factors *belief* and *reality congruency* and the between-subject covariate *initial-belief* (i.e., whether the TB block preceded the FB block or vice versa). We initially preregistered ANCOVAs, but since our covariate *initial-belief* was not metric, we ran ANOVAs instead. In accordance with the analysis of the first two trials only, the ANOVA yielded moderate evidence for a main effect of *reality congruency* (BF$_{10}$ = 5, see Table 3 for all Bayes Factors and model comparisons) and moderate evidence against a main effect of *belief* (BF$_{10}$ = 0.14). Again, participants were faster in reality congruent trials compared to reality incongruent trials (RC: Avg = 412 ms, SD = 96 ms, SE = 9 ms, RI: Avg = 421 ms, SD = 97 ms, SE = 9 ms, see also Fig 4 upper right). Contrary to our hypothesis but in line with experiment 1, analysis 2 provided moderate evidence against an interaction between *belief* and *reality congruency* (BF$_{10}$ = 0.13). There was, however, moderate evidence for a main effect of *initial-belief* (BF$_{10}$ = 4). The factor *initial-belief* was added as a covariate in this analysis characterizing whether a participant had started the experiment with the true belief condition (i.e., the transparent mask) or with the false belief condition (i.e., the opaque mask). Participants who started the experiment with the false belief condition were, on average, faster than participants who started with the true belief condition (Start with TB: Avg = 435 ms, SD = 118 ms, SE = 16 ms, Start with FB: Avg = 402 ms, SD = 71 ms, SE = 10 ms,

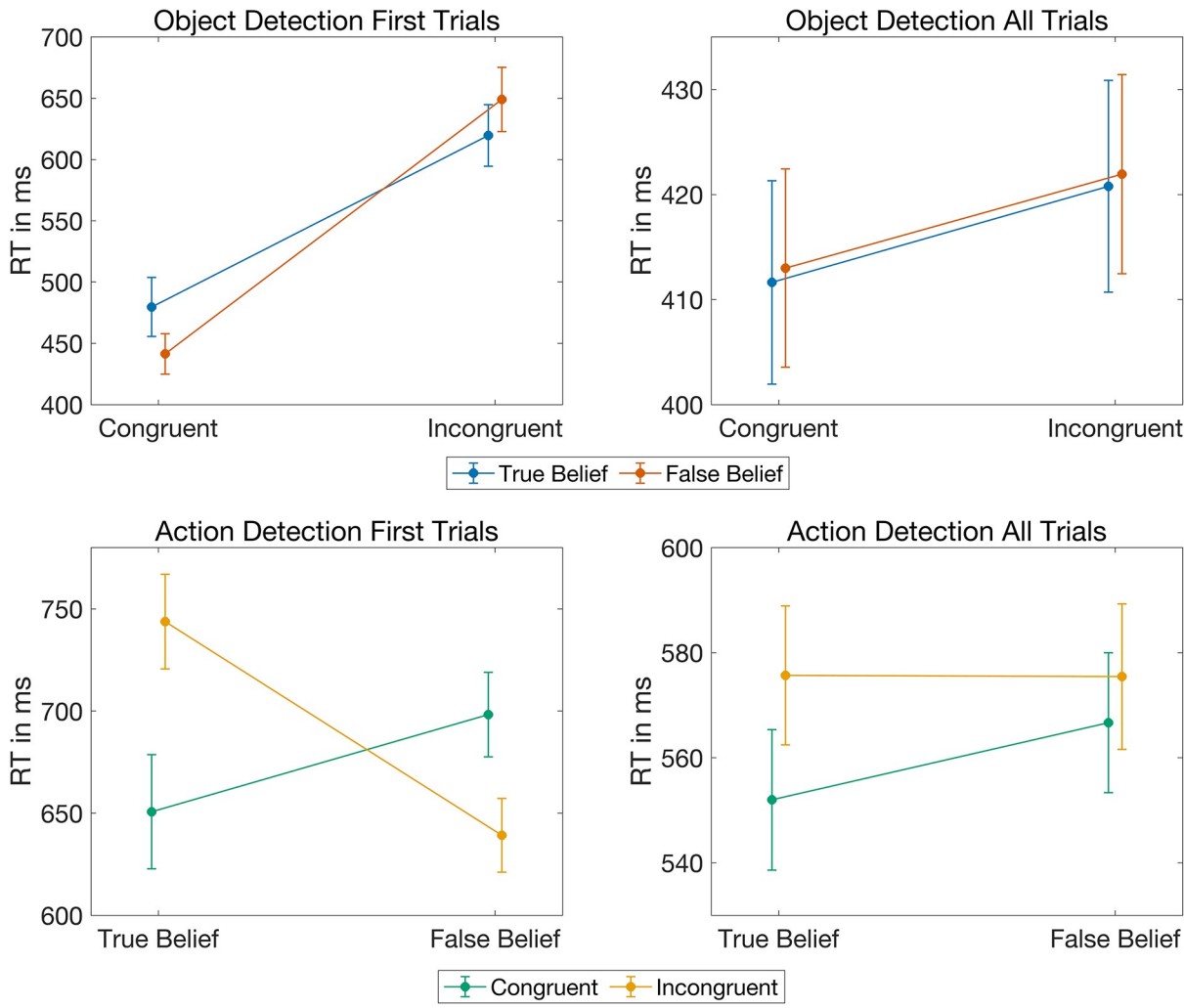

**Fig 4. Response times in experiment 2.** Upper left: Grand averages of the reaction times in the initial two trials of the object detection task along with standard errors for true belief (blue) and false belief trials (red) as a function of reality congruency; upper right: Grand averages of all reaction times collected in the lab for the object detection task along with standard errors for true belief (blue) and false belief trials (red) as a function of reality congruency; lower left: Grand averages of the reaction times in the initial two trials of the action detection task along with standard errors for reality congruent (green) and reality incongruent trials (orange) as a function of belief; lower right: Grand averages of all reaction times collected in the lab for the action detection task along with standard errors for reality congruent (green) and reality incongruent trials (orange) as a function of belief.

see Fig 5). Moreover, there was extreme evidence for an interaction between *initial-belief* and *belief* ($BF_{10} > 1,000,000$, see Fig 5 left graphic). That is, if participants started with the true belief condition, they responded faster in false belief trials than in true belief trials (TB: Avg = 449 ms, SD = 119 ms, SE = 16 ms, FB: Avg = 421 ms, SD = 121 ms, SE = 17 ms). Whereas, if they started with the false belief condition, they responded faster in the true belief trials (TB: Avg = 389 ms, SD = 74 ms, SE = 9 ms, FB: Avg = 415 ms, SD = 73 ms, SE = 10 ms), reflecting their decreased response times as a result of their experience with the task. Post-hoc paired t-tests confirmed the extreme evidence for a difference between true belief and false belief trials for both groups (for participants who started with the true belief condition: $BF_{10} > 1,000,000$, error = ±1.1%, for participants who started with the false belief condition: $BF_{10} > 1,000,000$, error = ±2.04%). Finally, the data provided moderate evidence against an interaction between *initial-belief* and *reality congruency* ($BF_{10} = 0.13$). Analogous to the above and

**Table 4. Bayes factors for the pairwise comparisons in the lab experiment.**

| | Task | Condition | Comparison | BF10 | Error in % |
|---|---|---|---|---|---|
| First Trials | Object Detection | RI | FB<TB | 0.148 | 6.209E-04 |
| | | RC | TB<FB | 0.077 | 0.001 |
| | Action Detection | FB | RI<RC | 3.631 | 1.12E-04 |
| | | TB | RC<RI | 31.198 | 2.73E-05 |
| All Trials | Object Detection | RI | FB<TB | 0.077 | 4.258E-05 |
| | | RC | TB<FB | 0.166 | 2.41E-04 |
| | Action Detection | FB | RI<RC | 0.047 | 7.89E-06 |
| | | TB | RC<RI | 2142.7 | 1.02E-08 |

Table 4 shows the Bayes factors along with the corresponding errors for the pairwise comparisons in the object detection and the action detection tasks for reality incongruent trials (RI, preregistered direction: reaction times in false belief trials are shorter than in true belief trials) and for reality congruent trials (RC, preregistered direction: reaction times in true belief trials are shorter than in false belief trials) in the analysis of the first two trials only and the analysis of all trials.

previous research findings [2, 3, 16], we also conducted a planned comparison including only the mean reaction times of reality incongruent trials (directed Bayesian t-test FB < TB). In line with the above results but contrary to prior findings without a blindfold, this test yielded strong evidence against participants being faster in reality incongruent trials when the agent had a false belief compared to when the agent had a true belief (BF$_{10}$ = 0.07, error < 0.0001%, see Table 4). Contrasting only the mean reaction times of reality congruent trials also revealed moderate evidence against an effect of belief (directed Bayesian t-test with TB < FB: BF$_{10}$ = 0.2, error < 0.0001%). Including the factor *initial-belief* in these comparisons did not alter the results.

**Action detection task–first trials.** As done for the object detection task, we only included the first two trials of the action detection task in a first analysis for comparison with the online experiment. To test for an egocentric bias in reaction times, we analyzed these initial responses using a linear mixed model including the within-subject factor *reality congruency*, and the

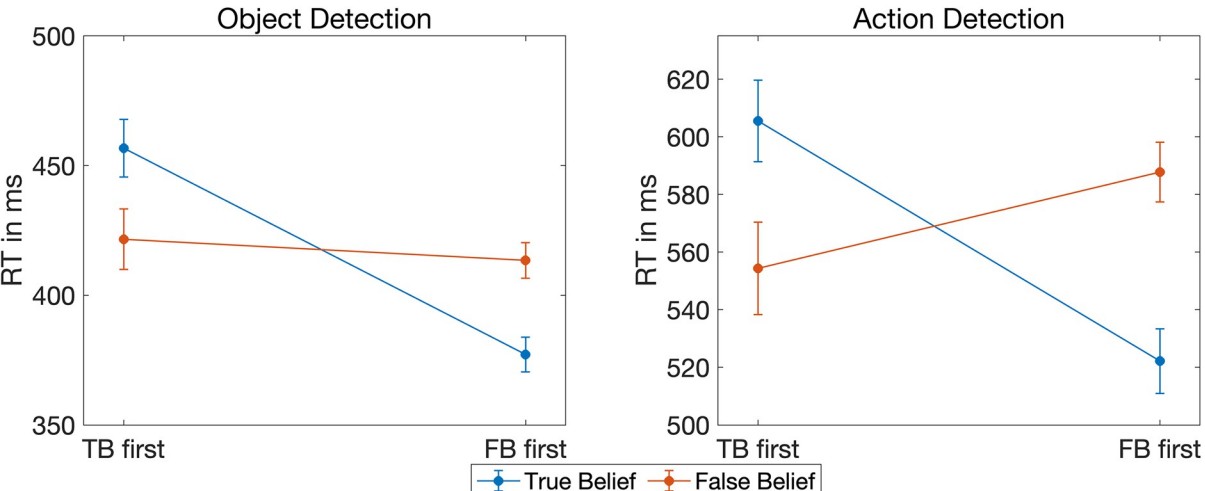

**Fig 5. Response times as a function of initial belief.** Left: Grand averages of the reaction times in the object detection task for participants who started the experiment with the true belief condition (i.e., the transparent glasses) and participants who started the experiment with the false belief condition (i.e., the opaque glasses). Right: Grand averages of the reaction times in the action detection task for participants who started the experiment with the true belief condition and participants who started the experiment with the false belief condition.

between-subject factor *belief*. As in experiment 1, this analysis yielded moderate evidence against both main effects (belief: $BF_{10} = 0.22$, reality congruency: $BF_{10} = 0.20$, see Table 3 for the Bayes Factors of all model comparisons), but extreme evidence for an interaction between the two factors ($BF_{10} = 387$). In true belief trials, participants responded faster if the agent reached into the box where the object actually was (i.e., reality congruent) instead of the empty box (i.e., reality incongruent) (RC: Avg = 651 ms, SD = 203 ms, SE = 28 ms, RI: Avg = 744 ms, SD = 152 ms, SE = 23 ms, see Fig 4 lower left). In false belief trials, in turn, participants responded faster if the agent reached into the empty box instead of the box where the object was actually located, in line with her belief (RC: Avg = 698 ms, SD = 151 ms, SE = 21 ms, RI: Avg = 639 ms, SD = 134 ms, SE = 18 ms). As in experiment 1, we preregistered two planned comparisons for the action detection task, one including only true belief trials (paired directed Bayesian t-test with RC < RI), and one including only false belief trials (paired directed Bayesian t-test with RI < RC). These comparisons revealed very strong evidence for an effect in true belief trials ($BF_{10} = 31$, error < 0.00003%, see Table 4), but only moderate evidence for an effect in false belief trials ($BF_{10} = 4$, error < 0.0001%).

**Action detection task–all trials.**   In a second analysis, we included all responses that were collected in the lab during the action detection task and conducted a repeated-measures ANOVA with the two within-subject factors *belief* and *reality congruency* and the between-subject covariate *initial-belief* to test for an egocentric bias in response times. As analysis 1, this ANOVA provided moderate evidence against a main effect of belief ($BF_{10} = 0.18$). Unlike analysis 1, however, the ANOVA yielded moderate evidence for a main effect of *reality congruency* ($BF_{10} = 3$), i.e., participants responded faster if the agent reached into the box with the object instead of into the empty box (RC: Avg = 559 ms, SD = 131 ms, SE = 12 ms, RI: Avg = 576 ms, SD = 133 ms, SE = 13 ms). The interaction between *belief* and *reality congruency* remained inconclusive ($BF_{10} = 0.51$). Thus, at the beginning of the experiment, participants' responses are mainly driven by the belief of the agent. Over the course of the experiment, however, reality congruency becomes the main driver of effects. The Bayes Factor of the main effect of the factor *initial-belief* remained inconclusive ($BF_{10} = 0.49$), but, as in the object detection task, there was extreme evidence for an interaction between *initial-belief* and *belief* ($BF_{10} > 1,000,000$). That is, participants who started with the true belief trials were faster in false belief trials (TB: Avg = 606 ms, SD = 145 ms, SE = 20 ms, FB: Avg = 554 ms, SD = 166 ms, SE = 22 ms, see Fig 5 left graphic). Conversely, participants who started with the false belief trials, were faster in the true belief trials (TB: Avg = 522 ms, SD = 115 ms, SE = 15 ms, RI: Avg = 588 ms, SD = 106 ms, SE = 14 ms), reflecting speeding up of the participants with increased experience across the trials. For the interaction between *initial-belief* and *reality congruency* the model comparison yielded moderate evidence against an effect ($BF_{10} = 0.16$).

As in the analysis of the first two trials only, we also conducted two planned comparisons in the analysis of all trials, one including only true belief trials (paired directed Bayesian t-test with RC < RI), and one including only false belief trials (paired directed Bayesian t-test with RI < RC). For true belief trials, we found extreme evidence for an effect of reality congruency ($BF_{10} = 2142$, error < 0.0001%), and strong evidence against such an effect in false belief trials ($BF_{10} = 0.047$, error < 0.0001%). Including *initial-belief* in these comparisons did not alter the results.

**Relation between altercentric and egocentric biases.**   In addition to our preregistered analyses and analogous to experiment 1, we examined the relation between the altercentric and the egocentric bias with a Bayesian correlation analysis. As in experiment 1, this analysis provided moderate evidence against a correlation between the altercentric and the egocentric bias (Pearson's r = -0.046, $BF_{10} = 0.183$).

## Discussion experiment 2 (lab)

The results of the lab experiment confirmed the results found in the online experiment. For the object detection task, there was evidence for a main effect of *reality congruency*. That is, participants responded faster if the object appeared where it was last seen. This effect was most pronounced in the first trials and became smaller over time. In none of the analyses was the expected altercentric bias found. That is, participants did not benefit from the false belief of the agent when the object appeared at the unexpected location, as reported in previous studies that did not use a blindfold [2, 3, 16]. Further, neither an effect of *belief* nor an interaction between *belief* and *reality congruency* were observed in the initial trials or across the whole experiment.

In contrast, participants' responses indicated that they processed the belief of the agent during the action detection task. Especially at the beginning of the task, participants' responses were mainly driven by their expectation that the agent would act according to her belief, which led to the expected crossing interaction between *belief* and *reality congruency*. The effect of *reality congruency* was less pronounced when the agent had a false belief as opposed to a true belief about the object's location, which already indicated egocentric interference. This egocentric bias became dominant across the entire experiment. Namely, the evidence for a crossing interaction between *reality congruency* and *belief* vanished (inconclusive Bayes Factor) when averaging across trials of the entire experiment. Instead, moderate evidence for a main effect of *reality congruency* was found. Further, there was still moderate evidence against a main effect of *belief*. These results suggest that, over time, participants were no longer impacted by the agent's belief. It is likely that participants realized that computing the agent's belief was not beneficial for the task, because she acted incongruently to her belief in half of the trials. Thus, their own perspective took over, leading to an increased egocentric bias over trials.

For both the object and the action detection task, we found extreme evidence for an interaction between the blindfold that participants experienced first and the agent's belief. Namely, participants were faster with the blindfold and corresponding belief that they experienced in the second block, presumably reflecting learning effects with the task. For the object detection task, the order in which participants experienced the blindfold also had an effect: Participants responded faster if they started the experiment with the false belief condition, i.e., with the agent wearing the opaque blindfold. One possible explanation for this finding could be that, knowing that the agent could not see anything in the first block of the experiment, might have made it easier for participants to ignore the agent's perspective throughout the rest of the experiment. Thus, the agent's ignorance in the beginning of the experiment might have helped participants to focus solely on their own perspective.

Finally, a correlation analysis provided moderate evidence against a linear relationship between the altercentric and the egocentric bias. Again, this null finding needs to be interpreted with caution as we did not find an altercentric bias.

## General discussion

Across two experiments with different settings and a large population pool, we found evidence that participants' reaction times in detecting an object were mainly driven by reality congruency and not modulated by the belief of an observer. That is, participants were a) faster to detect an object when its location was known, and b) were not influenced by the belief of a blindfolded bystander whose perspective they had to derive from their own experience. Thus, in contrast to previous studies where the bystander's belief had been manipulated through their presence or absence [2, 3, 9], we did not find an altercentric bias in participants' object detection. Since altercentric biases have been suggested to reflect implicit belief processing,

this result can be seen in the context of previous studies that reported replication difficulties for implicit ToM tasks, even when no blindfold manipulation was used [31–35].

When participants had to detect the agent's action toward the object, they were guided by the agent's belief. However, they were also influenced by their own knowledge of where the object should be, in form of an egocentric bias in their response times. These results are in line with other studies revealing egocentric biases in visual perspective taking [8, 36] and belief processing [25]. For instance, in the explicit version of the red dot visual perspective taking task introduced by Samson and colleagues [8], participants needed to report how many dots an agent within a visual scene could see. The agent saw either fewer or the same number of dots than the participants. Participants' responses were slower and less accurate when they could see more dots than the agent. Similarly, participants were less accurate in determining the agent's belief about an object's location when they themselves knew that the object was in a different location [25]. In the present task, the egocentric bias increased across trials reflecting the fact that, over time, the participant's own perspective became dominant over the agent's belief. This was likely due to the fact that the agent's belief was not a reliable indicator for her action, which participants might have realized in the course of the trials. In contrast, they did not ignore their own belief about the object's location, although it was not beneficial to the task either (namely, the agent also acted incongruent to the participants' belief in half of the trials). Thus, in case of unreliable information, individuals seem to favor their own perspective over that of someone else.

The fact that we were able to replicate two well-established cognitive phenomena, the reality congruency effect and the egocentric bias, suggests that our data are meaningful and strengthens our confidence in the absence of an altercentric bias in the current task. But what could be the reason for the lack of altercentric interference in the current experimental setting? Altercentric biases have been proposed as a measure of implicit belief processing. There are several other implicit ToM tasks that, for instance, use looking behavior as an indicator for action anticipation. Recently, replication difficulties have been reported for some of these tasks [e.g., 32, 37, 38]. Thus, in the context of these failed replications, the absence of an altercentric bias in the current task could be seen as an indicator that altercentric biases, just like implicit belief processing in general, are rather fragile phenomena that are difficult to tap. However, there were also important differences between our own and previous altercentric bias tasks [31, 32]. Unlike the original paradigm [2, 3, 16], we manipulated the location of the object rather than its presence. This may require a richer belief attribution, as a particular location needs to be ascribed to the agent's belief instead of the mere presence or absence of the object. Also, our participants completed a two-alternative forced-choice task, which may have been more challenging than a simple go/no-go task. Yet, it is unlikely that these differences explain our null results, as previous studies found altercentric effects with similar manipulations [9, 16].

More importantly, in the current study, participants had to infer the belief of the agent from their own experience with a blindfold. This blindfold manipulation was proposed as a critical test, since manipulating the agent's belief with their presence/absence or gaze direction allows for alternative explanations involving domain-general perceptual processes or lower-level social cognitive mechanisms that do not involve true mentalizing [e.g., 14, 15]. For instance, the agent's gaze direction could have functioned as an attentional cue in the original visual perspective taking studies [8]. Similarly, the agent's presence or absence during the location change in a false belief task could have had an impact on the saliency of the event and, thus, affected participants' memory of it, leading to the response time differences in previous studies [2, 3]. The blindfold manipulation excludes these explanations because all variables (e.g., visual input, attention orientation etc.) can be kept exactly the same, independently of the belief manipulation [e.g., 17–19].

Whereas the blindfold manipulation has already been applied in visual perspective taking studies, it had not been utilized to study altercentric biases in adult belief processing before. Although visual perspective taking and belief reasoning are two closely related abilities, visual perspective taking may occur mainly on the perceptual level, while beliefs cannot directly be read from an agent but need to be inferred cognitively. Thus, one interpretation of our finding of the lack of an altercentric bias in a blindfold false belief setting may be that the altercentric biases reported in previous studies may not reflect true mentalizing, but rather result from socio-attentional processes.

Another interpretation of our findings might be that inferring the belief of an agent by their mere presence or absence is cognitively less demanding than deriving the belief of a person from previous self-experience. Thus, the blindfold setting in the current study might have made too high cognitive demands for an altercentric bias to occur spontaneously. To infer the belief of the agent, participants had to recognize the blindfold, retrieve their own experience with it from memory, transfer this experience to the agent and derive what she can or cannot see to finally infer her belief about the object's location. Parts of this complex process, such as deriving whether an agent can or cannot see based on previous experience with a blindfold, have been shown to occur on an automatic level in the simpler context of gaze following [39]. However, in the context of false belief situations, the process as a whole may have been too complex to function spontaneously. This argumentation is in line with findings that high workload disrupts implicit false belief processing [40]. Differences in cognitive load could also explain the mixed results of the visual perspective taking studies discussed in the introduction as these had applied different kinds of blindfold manipulations that differed substantially with respect to their complexity, saliency, and implementation. For instance, a complex cloaking device was utilized in one experiment that only made dots of a specific color disappear [22], while all other experiments used goggles that were either transparent or opaque but differed considerably with respect to their saliency. This argumentation is consistent with recent studies showing that the saliency and relevance of the agent's perspective is crucial for the manifestation of altercentric intrusions [41–43]. In addition, some studies chose a block design for the belief manipulation, whereas in other studies, participants needed to keep track of the transparency of the blindfold on a trial-by-trial basis [20–22]. Deriving the agent's perspective from previous experience therefore made different demands in the different studies.

In sum, it appears that while our own perspective intrudes with our judgement of another person's actions, intrusions from the perspective and belief of others are less evident. We found no altercentric bias in object detection when the belief of the other person had to be inferred from self-experience. Furthermore, the other's perspective was even ignored after a while when it proved unreliable for predicting the agent's action. Finally, this is also supported by the finding that participants were overall faster at detecting the ball when the agent initially had a false belief, indicating that a false belief of the agent may have helped to fully ignore the others' perspective.

## Conclusion

Across two experiments and a large number of participants, we found evidence against an altercentric bias in an object detection task in a setting where the belief of a person had to be inferred from personal experience with a blindfold that the agent wore. Participants' reaction times were only driven by reality congruency, that is, they were faster to detect the object if it was revealed where they could expect it to be. Altercentric biases have been proposed to reflect implicit belief processing. Against this background, the lack of altercentric intrusions in the current task may indicate that participants did not implicitly process the belief of the agent.

Possibly, inferring the belief from one's own experience with the blindfold was cognitively too demanding to occur automatically, even when the agent's belief was task-irrelevant. When the task was to detect the agent's actions, in contrast, participant's response times were modulated by the agent's belief, evidencing that they were able to infer her beliefs when likely to be task-relevant. In the action detection task, in addition, participants' responses were modulated by their own knowledge about where the object really was, replicating an egocentric bias previously observed in visual perspective taking and ToM tasks [8, 25, 36]. Taken together, the results of the two tasks emphasize that the degree to which participants' performance is affected by the belief of others depends on whether this belief is relevant to their task. Moreover, it may also depend on how readily available the others' perspective is. That is, we may spontaneously be influenced by the perspective of others if it can be grasped at an immediate perceptual level, such as in gaze following or visual perspective taking, but may not take their perspective or belief into account if it needs to be inferred in a cognitively more effortful way, for example, from one's own experience about visual access.

## Supporting information

**S1 File.**
(DOCX)

## Acknowledgments

We thank Cornelia Henschel for her contribution to data collection.

## Author Contributions

**Conceptualization:** Katrin Rothmaler, Charlotte Grosse Wiesmann.

**Data curation:** Katrin Rothmaler.

**Formal analysis:** Katrin Rothmaler.

**Funding acquisition:** Charlotte Grosse Wiesmann.

**Investigation:** Katrin Rothmaler.

**Methodology:** Katrin Rothmaler, Charlotte Grosse Wiesmann.

**Project administration:** Katrin Rothmaler.

**Resources:** Charlotte Grosse Wiesmann.

**Software:** Katrin Rothmaler.

**Supervision:** Charlotte Grosse Wiesmann.

**Validation:** Katrin Rothmaler.

**Visualization:** Katrin Rothmaler.

**Writing – original draft:** Katrin Rothmaler.

**Writing – review & editing:** Katrin Rothmaler, Charlotte Grosse Wiesmann.

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
