## [Decision Letter · Decision Letter 0]

12 Jul 2023

PONE-D-23-03185Evidence against implicit belief processing in a blindfold taskPLOS ONE

Dear Dr. Rothmaler,

Thank you for submitting your manuscript to PLOS ONE. After careful consideration, we feel that it has merit but does not fully meet PLOS ONE’s publication criteria as it currently stands. Therefore, we invite you to submit a revised version of the manuscript that addresses the points raised during the review process.

The reviewers liked the paper and found the authors work and claims valuable and persuasive. However, one reviewer suggested considering an additional work and improving the discussion in light of the same.

We look forward to receiving your revised manuscript.

Kind regards,

Shrisha Rao, Ph.D.

Academic Editor

PLOS ONE

Journal Requirements:

3. We note that Figure 1 includes an image of a [patient / participant / in the study].

Reviewers' comments:

Reviewer's Responses to Questions

**Comments to the Author**

1. Is the manuscript technically sound, and do the data support the conclusions?

Reviewer #1: Yes

Reviewer #2: Yes

2. Has the statistical analysis been performed appropriately and rigorously? 

Reviewer #1: Yes

Reviewer #2: Yes

3. Have the authors made all data underlying the findings in their manuscript fully available?

Reviewer #1: Yes

Reviewer #2: Yes

4. Is the manuscript presented in an intelligible fashion and written in standard English?

Reviewer #1: Yes

Reviewer #2: Yes

5. Review Comments to the Author

Reviewer #1: Summary: In this work, the authors want to see if humans exhibit altercentric bias. They study this through a modification of existing object detection task, by introducing a blindfold (which may or may not be opaque). This ensures the stimuli remains the same, which was not so in earlier work.The participants are familiarized with the blindfolds, and the agent's belief could therefore, be inferred based on participants' experience. The second study is to understand the relationship between egocentric bias and altercentric bias (the two are expected to be opposing). To do this, they employ an action detection task. They find that reaction times were driven by congruency (not belief; therefore no altercentric bias).

The authors have pre-registered the experiments, and the statistical analyses look fine to me.

The complete data is NOT yet available, but I see that the authors have stated they'd make it available on OSF upon acceptance.

I am not sure whether there is enough of a contribution or novelty here, but I don't have any issues with the experiments, analyses, or conclusion.

Reviewer #2: This paper addresses a central question about implicit theory of mind by addressing the existence of an altercentric bias crated by another person's false belief. The two experiments are very impressive and the data carefully analysed. Very impressive.

The main objective is to sharpen claims based on existing evidence. For this the authors combined the method pioneered by Kovacs et al (2010) with transparent/opaque goggles. An effect with this addition would show that the bias could not be due to the socio-attentional processes but must be due to a cognitively inferred belief. Indeed if the altercercentric bias could have been found under these conditions it would have been valuable evidence for its existence. But the evidence is strongly negative.

This creates a different problem of interpretation as the authors acknowledge in their General Discussion. One of these counterarguments is that inferring a person's visual access from which goggles the person is wearing is beyond implicit processing. So no wonder that no altercentric bias could be found under these conditions. This counterargument could be attenuated if there was evidence that implicit processing is possible with goggles. There was, I think to remember, such evidence in the context of automatic gaze following (Teufel et al 2013). The authors might want to sharpen their conclusions with the help of this work—provided no strong counterevidence has accumulated since.

Teufel, C., Alexis, D. M., Clayton, N. S., & Davis, G. (2010). Mental-state attribution drives rapid, reflexive gaze following. Attention, Perception, & Psychophysics, 72(3), 695-705.

6. PLOS authors have the option to publish the peer review history of their article (what does this mean?). If published, this will include your full peer review and any attached files.

Reviewer #1: No

Reviewer #2: No

---

## [Author Response · Author response to Decision Letter 0]

19 Oct 2023

Response to the editor

We have ensured that the manuscript meets all style requirements.

We will ensure to provide repository information for our data upon acceptance.

3. We note that Figure 1 includes an image of a [patient / participant / in the study].

Thank you for this note. We have gathered the signed consent form and amended a statement that the individual in the figure has given written informed consent (as outlined in PLOS consent form) to publish her photo.

We have reviewed the reference list and added the following reference due to the comment of reviewer #2: Teufel, C., Alexis, D. M., Clayton, N. S., & Davis, G. (2010). Mental-state attribution drives rapid, reflexive gaze following. Attention, Perception, & Psychophysics, 72(3), 695-705.

Response to reviewer #1:

Reviewer #1: Summary: In this work, the authors want to see if humans exhibit altercentric bias. They study this through a modification of existing object detection task, by introducing a blindfold (which may or may not be opaque). This ensures the stimuli remains the same, which was not so in earlier work.The participants are familiarized with the blindfolds, and the agent's belief could therefore, be inferred based on participants' experience. The second study is to understand the relationship between egocentric bias and altercentric bias (the two are expected to be opposing). To do this, they employ an action detection task. They find that reaction times were driven by congruency (not belief; therefore no altercentric bias).

The authors have pre-registered the experiments, and the statistical analyses look fine to me.

The complete data is NOT yet available, but I see that the authors have stated they'd make it available on OSF upon acceptance.

I am not sure whether there is enough of a contribution or novelty here, but I don't have any issues with the experiments, analyses, or conclusion.

We thank the reviewer very much for their positive feedback.

Response to reviewer #2:

Reviewer #2: This paper addresses a central question about implicit theory of mind by addressing the existence of an altercentric bias crated by another person's false belief. The two experiments are very impressive and the data carefully analysed. Very impressive.

The main objective is to sharpen claims based on existing evidence. For this the authors combined the method pioneered by Kovacs et al (2010) with transparent/opaque goggles. An effect with this addition would show that the bias could not be due to the socio-attentional processes but must be due to a cognitively inferred belief. Indeed if the altercercentric bias could have been found under these conditions it would have been valuable evidence for its existence. But the evidence is strongly negative.

This creates a different problem of interpretation as the authors acknowledge in their General Discussion. One of these counterarguments is that inferring a person's visual access from which goggles the person is wearing is beyond implicit processing. So no wonder that no altercentric bias could be found under these conditions. This counterargument could be attenuated if there was evidence that implicit processing is possible with goggles. There was, I think to remember, such evidence in the context of automatic gaze following (Teufel et al 2013). The authors might want to sharpen their conclusions with the help of this work—provided no strong counterevidence has accumulated since.

Teufel, C., Alexis, D. M., Clayton, N. S., & Davis, G. (2010). Mental-state attribution drives rapid, reflexive gaze following. Attention, Perception, & Psychophysics, 72(3), 695-705.

We thank the reviewer very much for their enthusiastic feedback and their valuable input. As recommended, we included the argument in our discussion (page 32 line 730 and following):

Parts of this complex process, such as deriving whether an agent can or cannot see based on previous experience with a blindfold, have been shown to occur on an automatic level in the simpler context of gaze following [39]. However, in the context of false belief situations, the process as a whole may have been too complex to function spontaneously.

and also sharpened our conclusion (page 34 line 781-782):

That is, we may spontaneously be influenced by the perspective of others if it can be grasped at an immediate perceptual level, such as in gaze following or visual perspective taking, but may not take their perspective or belief into account if it needs to be inferred in a cognitively more effortful way, for example, from one’s own experience about visual access

We did not skip the entire argument, though, because we still believe that it is possible that inferring the belief of a person based on previous experience with a blindfold is cognitively more demanding than inferring whether or not a person can see through a blindfold in a simpler context like gaze following.

---

## [Editor Report · Decision Letter 1]

26 Oct 2023

Evidence against implicit belief processing in a blindfold task

PONE-D-23-03185R1

Dear Dr. Rothmaler,

We’re pleased to inform you that your manuscript has been judged scientifically suitable for publication and will be formally accepted for publication once it meets all outstanding technical requirements.

Kind regards,

Joydeep Bhattacharya

Academic Editor

PLOS ONE
---

## [Editor Report · Acceptance letter]

5 Nov 2023

PONE-D-23-03185R1 

Evidence against implicit belief processing in a blindfold task 

Dear Dr. Rothmaler:

I'm pleased to inform you that your manuscript has been deemed suitable for publication in PLOS ONE. Congratulations! Your manuscript is now with our production department. 

Kind regards, 

on behalf of

Dr. Joydeep Bhattacharya 

Academic Editor

PLOS ONE